# Bifurcation drives the evolution of assembly-line biosynthesis

Thomas J. Booth[1,3,6], Kenan A. J. Bozhüyük [1,4,5,6], Jonathon D. Liston [1], Sibyl F. D. Batey [1], Ernest Lacey [2] & Barrie Wilkinson [1✉]

Reprogramming biosynthetic assembly-lines is a topic of intense interest. This is unsurprising as the scaffolds of most antibiotics in current clinical use are produced by such pathways. The modular nature of assembly-lines provides a direct relationship between the sequence of enzymatic domains and the chemical structure of the product, but rational reprogramming efforts have been met with limited success. To gain greater insight into the design process, we wanted to examine how Nature creates assembly-lines and searched for biosynthetic pathways that might represent evolutionary transitions. By examining the biosynthesis of the anti-tubercular wollamides, we uncover how whole gene duplication and neofunctionalization can result in pathway bifurcation. We show that, in the case of the wollamide biosynthesis, neofunctionalization is initiated by intragenomic recombination. This pathway bifurcation leads to redundancy, providing the genetic robustness required to enable large structural changes during the evolution of antibiotic structures. Should the new product be non-functional, gene loss can restore the original genotype. However, if the new product confers an advantage, depreciation and eventual loss of the original gene creates a new linear pathway. This provides the blind watchmaker equivalent to the design, build, test cycle of synthetic biology.

[1] Department of Molecular Microbiology, John Innes Centre, Norwich NR4 7UH, UK. [2] Microbial Screening Technologies, Smithfield, NSW 2164, Australia. [3] Present address: School of Molecular Sciences, University of Western Australia, Crawley, WA 6009, Australia. [4] Present address: Molecular Biotechnology, Department of Biosciences, Goethe University Frankfurt, 60438 Frankfurt am Main, Germany. [5] Present address: Max-Planck-Institute for Terrestrial Microbiology, Department of Natural Products in Organismic Interactions, 35043 Marburg, Germany. [6] These authors contributed equally: Thomas J. Booth, Kenan A. J. Bozhüyük. ✉email: barrie.wilkinson@jic.ac.uk

Microbial natural products produced by modular biosynthetic assembly-lines, i.e. (type I) polyketide synthases (PKSs)[1] and non-ribosomal peptide synthetases (NRPSs)[2], are an important source of pharmaceutical and agrochemical agents. Examples include well known antibacterial molecules such as the polyketide insecticide spinosyn[3] and NRPS derived penicillins[4]. Importantly, biosynthetic assembly-lines provide thousands of natural product scaffolds, including many of our essential clinical agents.

Essentially, NRPS and PKS modular megasynth(et)ases give rise to highly functionalised biopolymers from a broad variety of monomers, referred to as extender units. Hundreds of extender units have been reported, typically derived from malonate in the case of PKSs[5] or amino acids in the case of NRPSs[6,7]. They are likened to assembly-line processes due to their hierarchical and modular structures in which multiple, repeating modules of enzymatic domains catalyse the incorporation of an extender unit into the growing chain, along with any programmed additional chemical modifications, before transferring the elongated chain to the next module. The archetypical minimal assembly-line module consists of three core domains. Firstly, a domain for the selection and activation of an extender unit, the acyltransferase (AT) domains for PKSs or adenylation (A) domains for NRPSs. The activated substrate is then covalently attached to a prosthetic phosphopantetheine group of a small acyl carrier protein (ACP; PKSs) or peptidyl carrier protein (PCP; NRPSs) domain. Finally, the ketosynthase (KS; PKSs) or condensation (C; NRPSs) domains then link the covalently bound substrates to the growing polyketide or peptide chain. Although the exact mechanisms and ancillary domains of PKSs and NRPSs differ, the fundamental principal is that modules condense covalently bound substrates in a linear fashion. The inherent logic of this mechanism means that there is a direct relationship between the sequence of domains in an assembly line and the chemical structure of the resulting molecule[1,8,9]. In principle this relationship enables the prediction of natural product chemical structures directly from DNA sequences. In turn, this logic has inspired efforts to rationally reprogramme assembly lines to produce tailor-made molecules.

Numerous examples of assembly-line engineering have been reported; however, many display productivities well below that of the parental (wild-type) system. Insights into structural flexibility, inter-domain communication and the role of proof-reading by catalytic domains pre-empted novel strategies to engineer assembly-line proteins[10–14]. There is also an increasing body of evidence suggesting that we might further improve our ability to engineer these systems if we had a better understanding of their evolution[1,15–18].

With this latter point in mind, we have been searching for biosynthetic pathways that may represent transitional evolutionary states and provide exemplar systems to inform future work. Natural selection acts upon phenotypes, yet even a small change to the structure of a natural product can have profound effects on bioactivity and, thus, the fitness of the producing organism. Therefore, we hypothesised that strains encoding BGCs evolving new functionalities might be expected to produce multiple related products (congeners) (Fig. 1a). Here, we describe the BGC encoding of the wollamide-desotamide family of antibiotics, which represents an evolutionary snapshot of a modular NRPS assembly-line.

The wollamides are cyclic hexapeptides (Fig. 1b) that were disclosed in 2014[19]. The only reported producer of the wollamides is *Streptomyces* sp. MST-115088[19] and they exhibit potent antimycobacterial activity that has attracted the attention of synthetic medicinal chemists[20–23]. Along with the wollamides, *Streptomyces* sp. MST-115088 produces a related group of hexapeptides, the desotamides[24]. The desotamides and wollamides share a common peptide scaffold, except for a single residue change from glycine in the desotamides to D-ornithine in the wollamides (Fig. 1b). It is important to note that the previously identified desotamide producer, *Streptomyces scopuliridis* SCSIO ZJ46, is not reported to produce wollamides and the desotamide (*dsa*) BGC follows canonical NRPS logic[25]. The NRPS is encoded by three genes (*dsaI*, *dsaH* and *dsaG*) encoding two modules each. Importantly, module six, encoded by DsaG, incorporates glycine as the final amino acid. The ability of *Streptomyces* sp. MST-115088 to produce congeners with D-ornithine and glycine in the same position is therefore difficult to rationalise as, under a canonical NRPS mechanism, A-domains are responsible for the selection and activation of specific amino acid substrates. While A-domains can activate structurally related amino acids giving rise to families of structurally similar congeners (for example the combinations of valine, leucine, or allo-leucine at positions 3 and 4 of the wollamides-desotamides), the ability of an A-domain to activate substrates with such varying physico-chemical properties as glycine and ornithine is without precedent. Therefore, to explain the production of wollamides and desotamides by a single strain we hypothesised three scenarios (Fig. 1a): dual specificity of module six for glycine and ornithine; duplicated genes encoding module six homologues, each specific to glycine or ornithine, respectively; or duplicated BGCs where, as above, the final modules of each BGC are specific to glycine or ornithine.

Herein, we describe our combined bioinformatics, in vivo engineering, and biochemical analysis demonstrating how the *dsa* BGC evolved from an ancestral wollamide producing BGC via the process of gene duplication and intragenomic recombination with a second NRPS encoding locus. This allows us to propose an update to the current model for the evolution of assembly-line BGCs through which duplication and the resulting bifurcation reduce selective pressure and drive the evolution of new functions.

## Results

**Desotamides and wollamides are produced by a bifurcated NRPS assembly-line.** The genome of *Streptomyces* sp. MST-115088 was sequenced using Pacific Biosciences RS2[26] single-molecule technology and assembled using the HGAP3[27] pipeline to generate a single 7.9 Mb chromosomal assembly (Supplementary Table 1) (GenBank: CP074380). Analysis of the assembly using antiSMASH v4.0[8] allowed us to rapidly identify the wollamide (*wol*) BGC (Supplementary Table 2), which was then compared to the desotamide (*dsa*) BGC previously reported from *Streptomyces scopuliridis* SCSIO ZJ46 (GenBank: KP769807) (Fig. 2a, b). Five additional desotamide producers (MST-70754, MST-71458, MST-71321, MST-94754 and MST-127221) were identified through metabolomic analysis of Microbial Screening Technologies' unique collection of more than fifty thousand Australian actinomycete strains (Supplementary Information). The desotamide BGCs (Supplementary Tables 3–7) of these strains display functionally identical architectures (GenBank: MZ093610, MZ093611, MZ093612, MZ093613, MZ093614) to that reported from *Streptomyces scopuliridis* SCSIO ZJ46[25] (>95% nucleotide identity) (Supplementary Fig. 1). The *wol* BGC displays a similar architecture to the *dsa* BGCs but contains two genes *wolG1* and *wolG2* that are duplicates of the *dsaG* gene that encodes the final two modules of the NRPS. Similarly, there are duplicated homologues of the *dsaF* gene (*wolF1* and *wolF2*), which encodes an MbtH-like protein involved in A-domain functionality[28,29]. The *wol* BGC also contains six additional genes *wolRSTUVW* whose gene products are predicted to be involved in the biosynthesis of L-ornithine consistent with the presence of ornithine in position 6 of the wollamides.

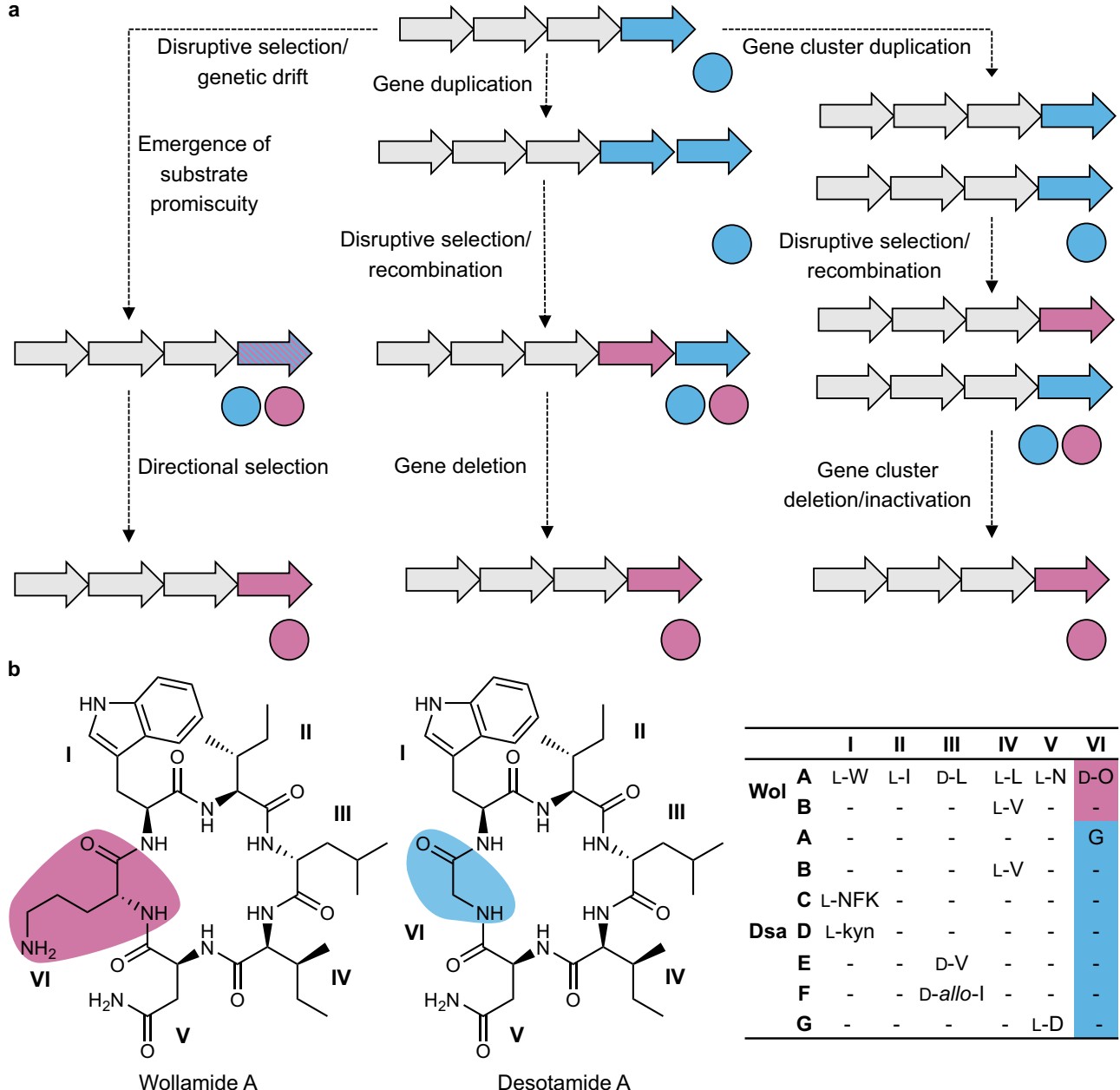

**Fig. 1 Intermediate chemotypes in the evolution of natural product biosynthetic gene clusters (BGCs). a** A cartoon demonstrating the three main models of gene cluster evolution depicting the transition of a fictitious BGC from chemotype A to chemotype B (coloured circles). Evolutionary processes are represented by dashed lines. **b** The structural diversity of the wollamides and desotamides. The ᴅ-ornithine and glycine residues, which define the wollamides and desotamides, respectively, are highlighted. The table on the right-hand side shows the variable positions of the various wollamide and desotamide congeners (NFK N-formyl kyunerine, Wol wollamide, Dsa desotamide).

During desotamide biosynthesis, DsaG is responsible for the final two rounds of peptide chain elongation. We hypothesised, therefore, that WolG1 and WolG2 may encode two forks of a bifurcated biosynthetic pathway where the first four rounds of peptide elongation proceed via the colinear activity of WolI and WoH, but the final two elongation steps are catalysed either by WolG1 or WolG2, yielding desotamide or wollamide products, respectively (Fig. 2c). Consistent with this hypothesis, in silico analysis of A-domain specificities[30,31] predicted the substrates for the final A-domains of WolG1 and WolG2 (henceforth WolG1A2 and WolG2A2) to be glycine and ʟ-ornithine (Supplementary Table 8).

**Engineering wollamide production in a desotamide-only producing strain.** To confirm our biosynthetic hypothesis and to explore the evolutionary relationship between the wollamide and desotamide pathways, we sought to engineer the co-production of wollamides into a desotamide-only producing strain through the heterologous expression of *wolG2*.

As NRPS genes are typically large (for reference, *wolG2* is 7.9 kb) and difficult to clone through conventional methods, we generated pBO1 (Supplementary Fig. 2), an integrative *Streptomyces* expression vector capable of propagating in both *Escherichia coli* and *Saccharomyces cerevisiae*. This allows larger genes to be assembled efficiently in yeast via transformation

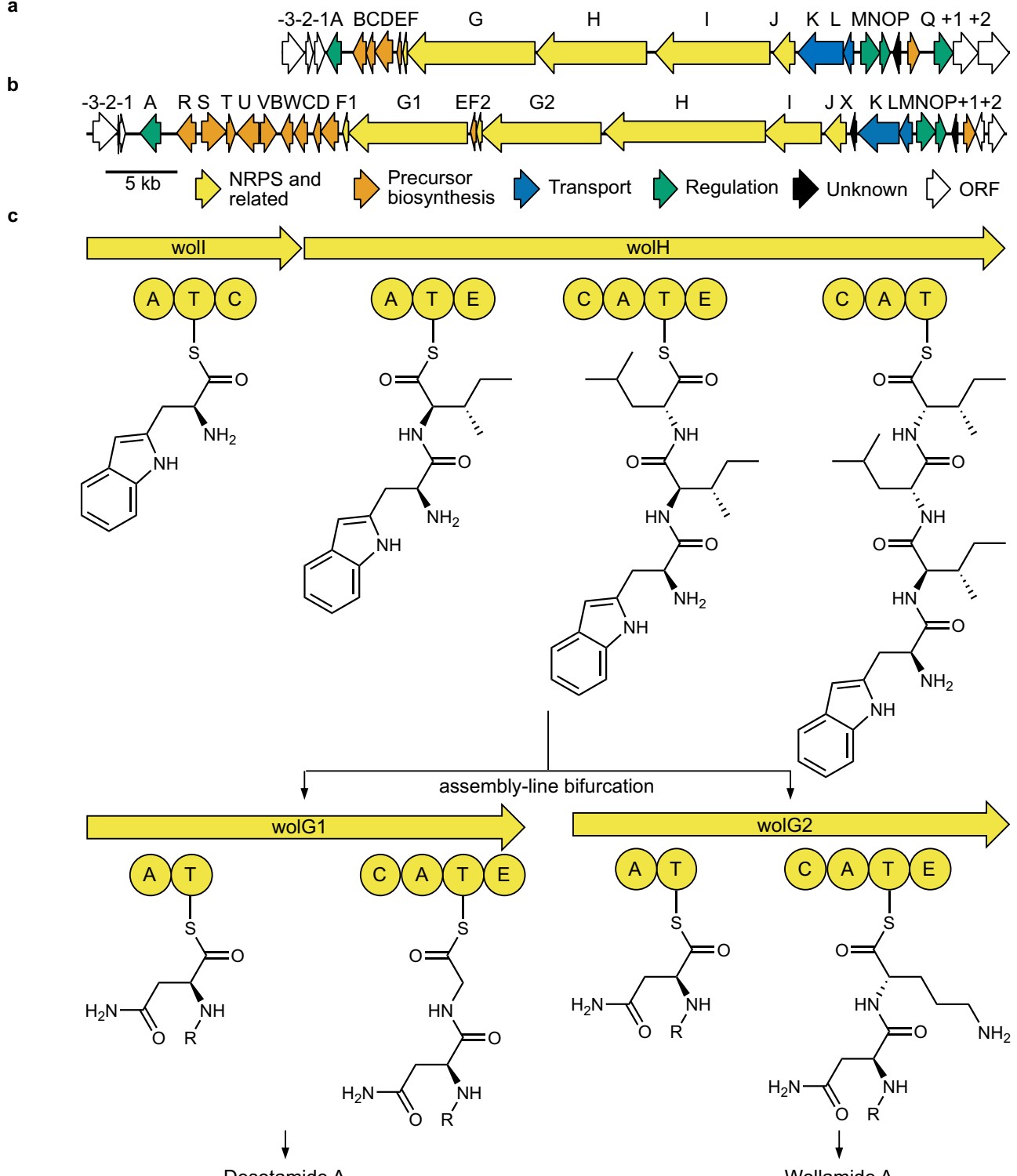

**Fig. 2 The genetic basis for bifurcated biosynthesis of desotamide and wollamide by *Streptomyces*. sp. MST-110588. a** The architecture of the desotamide producing *dsa* BGC from *Streptomyces scopuliridis* SCSIO ZJ46[25]. **b** The architecture of the wollamide and desotamide producing *wol* BGC from *Streptomyces* sp. MST110588. **c** The proposed biosynthetic pathway of the wollamides and desotamides. The first three condensations are catalysed by WolI and WolH. The final two condensations are catalysed by WolG1 to produce desotamides or WolG2 to produce wollamides.

associated homologous recombination[32] downstream of a constitutive promotor and subsequently transferred from *E. coli* to *Streptomyces* spp. through conjugal transfer (Supplementary Information). To assemble a *wolG2* expression plasmid, pBO1 along with target fragments amplified from genomic DNA were

transformed into *S. cerevisiae* CEN.PK 2-1C[33]. The resulting plasmid pBO1-*wolG2* was transformed into the desotamide producer *Streptomyces* sp. MST-70754[34] via conjugation.

*Streptomyces* sp. MST-70754 and its progeny carrying pBO1-*wolG2* were grown in triplicate under desotamide producing

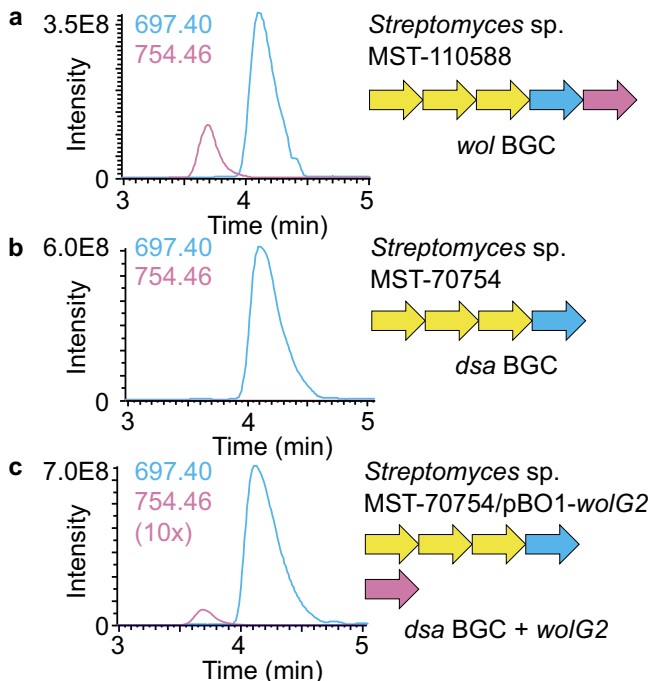

**Fig. 3 Heterologous production of the wollamides by overexpression of wolG2. a** The extracted ion chromatograms (EIC) for masses corresponding to desotamide A ([M + H] = 697.40) and wollamide A ([M + H] = 754.46) from wild-type *Streptomyces* sp. MST-110588 producing both compounds. **b** The EICs of desotamide A and wollamide A from *Streptomyces* sp. MST-70754 producing desotamide A only. **c** The EICs of desotamide A and wollamide A from *Streptomyces* sp. MST-70754/pBO1-wolG2 producing both compounds. Schematic representations of the assembly lines and overexpressed genes are also shown. (dsa desotamide, wol wollamide).

conditions and methanolic culture extracts were then analysed by LCMS in comparison to the wollamide producer *Streptomyces* sp. MST-115088 (Fig. 3). The presence of both wollamide and desotamide congeners was confirmed for the engineered strain by comparison of retention time, isotopic masses and MS/MS fragmentation of the compounds produced by the native wollamide producer (desotamide: $[M + H]^+ = 697.4047$, $[M + Na]^+ = 719.382$; wollamide A $[M + H]^+$ 754.4655, $[M + Na]^+ = 776.4419$) (Fig. 3) (Supplementary Figs. 3–5).

This engineering strategy relied on the assumption that the docking domains (DDs) of DsaH and WolG2 that mediate NRPS interactions would still function as a pair[35,36]. Production of wollamide A by *Streptomyces* sp. MST-70754/pBO1-wolG2 shows that this interaction is still possible; however, the yields are relatively low (approximately 50-fold lower than *Streptomyces* sp. MST-110588, Fig. 3c, Supplementary Fig. 3). Based on this observation, we hypothesised that some depreciation of the DD interaction may have occurred.

To investigate this, we compared the N-terminal DD primary sequences of WolG1 and WolG2 (Supplementary Fig. 6), as well as calculated homology models of the DDs in complex with the WolH DD (Supplementary Fig. 7). Alignment of the DDs revealed a glutamate to alanine mutation at position (E16A). Homology modelling with the previously characterised PaxB docking domain (6TRP_1)[37] revealed that the E16A amino acid change in WolG2 leads to the loss of a salt bridge formed with R3504 from WolH. Although other factors may play a role, the importance of this salt bridge in DD interactions suggests that

this mutation is likely to account for the observed differences in yield.

**Evolution of adenylation domain specificity via intragenomic recombination.** The unusual architecture of the *wol* BGC led us to consider potential mechanisms for its evolution. The high similarity of *wolG1* to *wolG2* (80.9%) and *wolG1/wolG2* to *dsaG* (74.3% and 71.2% respectively) is indicative of an ancestral gene duplication event[38]. Additionally, *wolG1*, *wolG2* and *dsaG* share common patterns of nucleotide skew, a phenomenon indicative of recent divergence. However, despite this high similarity, there is a notable drop in nucleotide identity within the region coding for the final adenylation domains which is also manifest in the gene products (54.6% nucleotide and 29.2% protein sequence similarity) (Supplementary Fig. 8). Given the overall similarity of these genes, we deemed it unlikely that such high sequence variation could emerge through the accumulation of point mutations alone. Furthermore, phylogenetic reconstructions of A-domains hinted at independent evolutionary histories when compared to the rest of the assembly line (Supplementary Figs. 9–11). Similar patterns have been observed in other bacterial NRPS clusters[18,39–41].

Many studies have highlighted the role recombination, e.g. the exchange of alleles via a double recombination event, plays in the evolution of assembly-lines[1,42]. More specifically, given the reduced rate of horizontal gene transfer between distantly related taxa and the high rate of heterogeneity between recombinant sequences, it has been speculated that intragenomic recombination within ancestral strains can provide opportunities for assembly-line diversification[18,43]. To assess the possible role of intragenomic recombination in the evolution of the *wol* BGC we generated a nucleotide sequence alignment of all thirty-four NRPS-associated adenylation domains present in the *Streptomyces* sp. MST110588 genome for analysis using the Recombination Detection Programme 4 (RDP4)[44].

Using our dataset, RDP4[44] predicted 29 potential intragenomic recombination events (Supplementary Fig. 12 and Supplementary Table 9), 11 of which were supported by two or more methods, allowing recombination breakpoints to be predicted. Only two recombination events were supported unanimously, including one event, in which *wolG1A2* was identified as a recombinant sequence with *wolG2A2* as the major parent and an adenylation-domain-encoding sequence from elsewhere in the genome, encoded by *orf6595A*, as the minor parent (Fig. 4a–c and Supplementary Table 9). The predicted recombinant region lies between the conserved motif A2 (N-terminal) and motifs A5 and A6 (C-terminal) of the A-domain, thus comprising the flavodoxin subdomain[18,39,41] and a large portion of the N-terminal subdomain (Fig. 4d). Crucially, this would allow for the substitution of the amino acid binding pocket and catalytic P-loop while maintaining the C-A-domain interface (Supplementary Fig. 13). This pattern is seen frequently in our dataset (Supplementary Fig. 12) and may suggest an advantage for maintaining the structural relationships between the P-loop and substrate binding pocket.

Importantly, the adenylation domain encoded by *orf6595* was predicted in silico to select for glycine (Supplementary Table 8) meaning that the predicted recombination event could theoretically convert a wollamide producing assembly line into a desotamide producer.

**Biochemical validation of A-domain substrate specificities.** To gain insight into the function of the minor parent BGC, the sequences of Orf6595 and proteins encoded by the surrounding

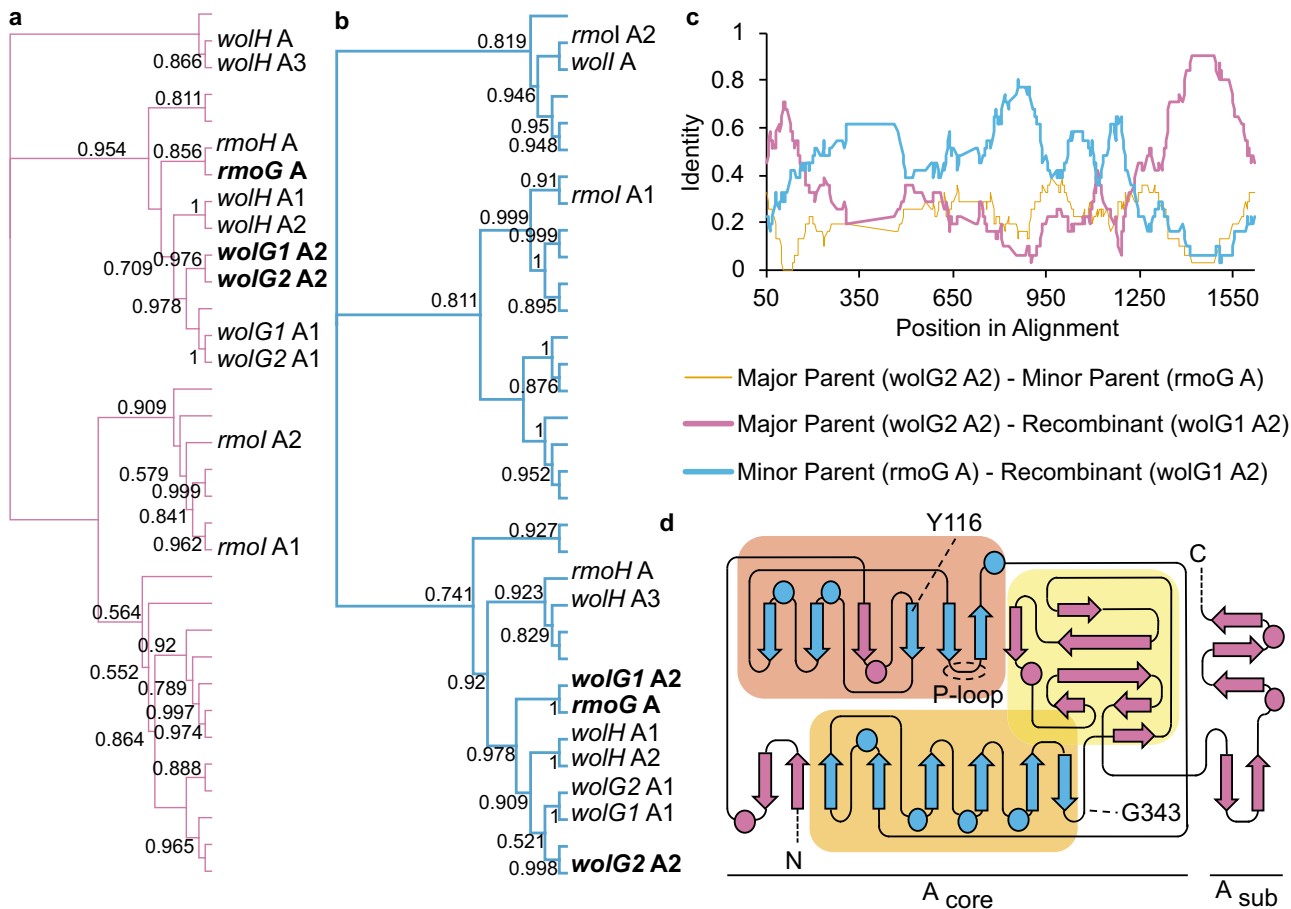

**Fig. 4 Evidence for the intragenomic recombination of adenylation domains. a** DNA phylogeny of the external region of the adenylation domains, showing the close relationship of the major parent, *wolG2* A2, and the recombinant, *wolG1* A2. **b** DNA phylogeny of the internal region of the adenylation domains, showing the close relationship of the minor parent, *rmoG* A, and the recombinant, *wolG1* A2. **c** Nucleotide identity between the three adenylation domains based on a 50 nucleotide sliding window. **d** Topology of an adenylation domain based on ref. [104], showing the predicted recombination sites Y116 and G343. Features are colour-coded according to panel **a**.

genes were searched against the MIBiG database[45] using cblaster[46]. The search identified the BGC as homologous to the rimosamide (*rmo*) BGC from *Streptomyces rimosus* ATCC 10970[47] (Supplementary Table 10). More specifically, Orf6595 is a homologue of RmoG which encodes a single NRPS module (C-A-T) that is known to incorporate glycine into the rimosamide peptide chain. This was consistent with bioinformatic predictions of the Orf6595 adenylation-domain active site providing strong evidence that recombination between the adenylation domains of *orf6595* and *wolG2* could confer specificity to glycine (Supplementary Fig. 14).

To verify this prediction and examine the substrate specificity of all of the A-domains of interest, pET28a hexa-histidine tagged WolG1A2, WolG2A2 and Orf6595A constructs were cloned for expression based upon the A-domain boundaries as described in Crüsemann et al.[39] These were expressed in *E. coli* Rosetta 2(DE3)pLysS and purified using Ni-Affinity chromatography. Initially, the resulting protein was insoluble; however, co-expression with the MbtH-like protein WolF2 (expressed from pCDFDuet-1) enabled the purification of soluble protein in each case. The ability of these isolated A-domains to activate each of the twenty proteinogenic amino acids and L-ornithine was then measured using a hydroxylamine trapping assay[48] (Supplementary Table 11). WolG2A2 adenylates L-ornithine, in line with our hypothesis (Fig. 5a); however, it was also capable of activating other substrates albeit with lower efficiency. Most noticeably,

WolG2A2 accepted L-aspartate (58% activity relative to L-ornithine) and L-asparagine (44% activity relative to L-ornithine) as substrates, but wollamide analogues in which aspartate or asparagine were substituted for ornithine were not identified in culture extracts of *Streptomyces* sp. MST-110588 despite targeted LCMS analysis. In contrast, both WolG1A2 and Orf6595A activate glycine in a highly specific manner (Fig. 5a). These data corroborate our hypothesis that historic recombination with *orf6595* could alter the substrate specificity of the module six adenylation domain of WolG2 from L-ornithine to glycine. To test this hypothesis further, we produced a hybrid A-domain encoding gene sequence representing the hypothetical ancestral recombinant, based on *wolG2A2* and *orf6595* sequences (Fig. 5b). The resulting gene product was purified and assayed as above and found to be highly selective for glycine (Fig. 5a).

Taken together with combined genomic, in silico and in vivo data above, these biochemical data show how the contemporary desotamide BGCs have evolved from an ancestral wollamide-like BGC through the process of gene duplication, intragenomic recombination and gene loss.

## Discussion

How NRPS and PKS assembly-line biosynthetic pathways evolve is a question of perennial interest in natural products research. An understanding of this process offers new avenues for developing rational approaches for bioengineering and the targeted

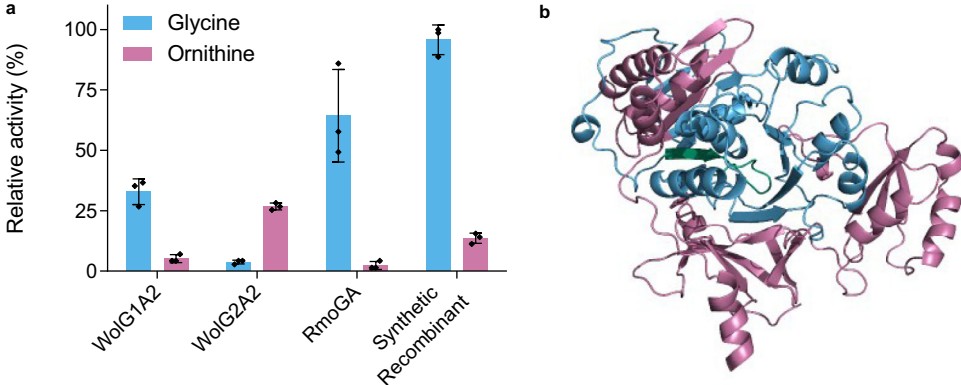

**Fig. 5 Substrate specificities of wild-type and recombinant adenylation domains. a** Relative activities of adenylation domains as determined by hydroxylamine trapping assays[47]. Each amino acid and enzyme combination was tested in triplicate ($n = 3$); the mean value was calculated and the standard deviation is shown. **b** Structural homology model of the synthetic recombinant adenylation domain. The model is colour-coded depending on the origin of the peptide sequence, pink for WolG2A2 and blue for RmoGA. The P-loop is highlighted in green.

production of new molecules. Evolutionary analyses of publicly available BGC sequences are common in the literature[18,49,50], but it is unprecedented to find an example of a BGC that represents a snapshot of assembly-line evolution. In this work, we studied *Streptomyces* sp. MST110588, a co-producer of desotamide and wollamide hexapeptide antibiotics. Despite their close structural relationship, it was not obvious how such a mixture of congeners might be assembled based on canonical NRPS function, pointing to an unusual biosynthetic pathway.

The architecture of the wollamide BGC is highly similar to the previously described desotamide BGC; however, a striking difference is that it contains duplicates (*wolG1* and *wolG2*) of *dsaG* which encodes the final NRPS protein of the desotamide assembly line. This suggested a bifurcated assembly-line was responsible for the observed mixture of desotamide and wollamide congeners (Fig. 2c). In this scenario, WolI and WolH are responsible for condensation of the first four amino acids prior to extension with either WolG1, producing desotamides, or WolG2, producing wollamides. This hypothesis was supported by bioinformatic (Fig. 4) and biochemical analysis (Fig. 5a) of the A-domains WolG1A2 and WolG2A2, which encode the selection and activation of glycine and L-ornithine respectively. Further in vivo evidence came from strain engineering in which expression of WolG2 in a desotamide-only producing strain led to the biosynthesis of additional wollamide congeners (Fig. 3).

Comparison of the protein sequences of the NRPSs encoded by *wolG1*/*G2* indicated a high degree of sequence conservation and identical GC-skews, indicative of a gene duplication event. The regions coding for the final adenylation domains showed a marked drop in identity. These data were indicative of a recombination event rather than of divergence through the accumulation of mutations alone, and we subsequently analysed all 34 A-domain sequences present in the *Streptomyces* sp. MST110588 genome using the recombination detection program RDP4[44]. This identified *wolG2* as the major parent, and an NRPS gene *orf6595* (encoded ~3 Mb away on the chromosome) as the minor parent, of a recombination event that formed *wolG1*. Using the cblaster tool we identified the BGC containing *orf6595* as a homologue of the rimosamide BGC[47]. Orf6595 is a homologue of RmoG that selects for glycine. This was verified by subsequent bioinformatics and biochemical analysis. To validate the recombination event predicted by RDP4 we used the contemporary sequences of *wolG2A2* and *orf6595* to recapitulate the predicted ancestor and generated a synthetic A-domain. Subsequent biochemical analysis of the isolated protein showed it was selective for glycine activation as predicted.

Based on these combined data we can confidently trace the evolutionary history of the wollamide and desotamide BGCs (Fig. 6a). First, a gene duplication event in an ancestral wollamide (or wollamide-like) BGC resulted in a redundant copy of the bimodular NRPS encoding modules 5 and 6 of the assembly line. Subsequently, an intragenomic recombination event between the DNA encoding the adenylation domains of the duplicated module six in the *wolG* homologue and *orf6595* resulted in an intermediate NRPS selective for glycine (leading to WolG1) and L-ornithine (leading to WolG2). This progenitor is the common ancestor of the wollamide and desotamide producing *wol* BGC sequenced here. In one lineage, selection and mutation resulted in the *wol* BGC observed in the contemporary genome of *Streptomyces* sp. MST110588 capable of bifurcated biosynthesis. In a divergent lineage, the ancestral gene encoding the L-ornithine-specific adenylation domain along with the duplicated MbtH-like protein-encoding gene and associated genes encoding L-ornithine biosynthesis were lost through gene deletion, ultimately resulting in the contemporary *dsa* BGCs producing only desotamides. These observations are consistent with the presence of a redundant epimerase domain located in the second modules of WolG1 and all DsaG homologues[51,52]. Moreover, in *Streptomyces* MST110588, production of wollamide congeners is approximately three times lower than that of desotamides. Our engineered strain of the desotamide producer, *Streptomyces* sp. MST-70754, expressing *wolG2* also showed an even larger difference in titer. This observation suggested that the interactions between the C- and N-terminal docking domains of WolH/DsaH and WolG2 had depreciated. To assess this, we calculated homology models which revealed that a key E16A amino acid change in WolG2 leads to the loss of a salt bridge formed with R3504 from WolH, most likely causing decreased DD pair affinities, explaining the observed differences in peptide titers. The depreciation of this interaction is consistent with a loss of selective pressure for wollamide production and may indicate drift towards a desotamide-only BGC. Although DD depreciation is likely to account for the difference in yield between the wollamides and desotamides in WT MST-110588, other factors may account for further loss of titre in our engineered strains, such as differences in the MbtH domains or the reduced capacity to biosynthesise L-ornithine.

Gene duplication and divergence is a central mechanism behind the evolution of new gene functions[38,53,54]. Gene duplications may occur as whole gene duplications (characterised by the duplication of an entire gene), or as partial or intragenic duplications (characterised by incomplete duplication of the

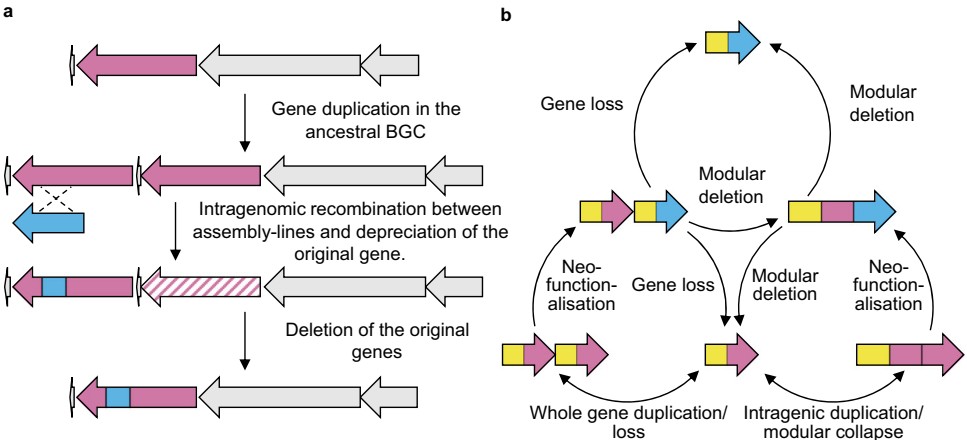

**Fig. 6 The evolution of biosynthetic assembly-lines. a** Proposed evolutionary history of the wollamide (*wol*) and desotamide (*dsa*) BGCs. **b** An updated model for the evolution of biosynthetic assembly-lines.

ancestral gene and can result in attenuated or lengthened duplicates, respectively[38]). There is convincing evidence for the role of intragenic duplications during assembly-line evolution (Fig. 6b). Such processes are often cited as the origin of multi-modularity[1,55] and there is significant evidence for this process, especially in PKS assembly-lines. In contrast, whole gene duplication has the potential to create bifurcated biosynthetic pathways, as evidenced here by the *wol* BGC, or parallel pathways, as evidenced by the recently published BE-18257A-C and pentaminomycins A-E BGC[56]. Through either mechanism, whole gene duplication creates redundancy and has the potential to reduce selective pressure for maintenance on both alleles[38,54]. In this fashion, a duplicate may evolve new functionality (neofunctionalisation) without the organism losing the original product. Thus, neofunctionalisation of one copy becomes possible without strong selective pressure against the loss of a phenotype and may follow a more gradual route for the emergence of a new activity. Subsequently, the original allele can be lost if there is no advantage and selection pressure for its maintenance leading to a new linear pathway (Fig. 6b). Moreover, this model provides a scenario where rounds of duplication and neofunctionalisation can occur but if no advantage is gained then the duplicated allele can be lost, restoring the original genotype. As genes and pathways are subject to multiple selective forces it is unclear how common this mechanism may be in nature. This is compounded by the homogenising effects of gene conversion/concerted evolution that can make it difficult to reliably distinguish duplication events[1,55,57,58]. However, the *wol* BGC provides compelling evidence that this process has occurred at least once, during the evolution of desotamide biosynthesis. A growing number of hexapeptide BGCs are becoming available, for example, those of the related ulleungmycin and curacomycin[59,60]. Given the small yet diverse differences of the peptide backbones, the hexapeptides present an enticing model for further understanding NRPS evolution.

Finally, the question arises as to whether our knowledge of the NRPS evolution can also be used to develop new engineering approaches. Recently, several efficient and/or highly productive engineering strategies have been published[61–64]. Yet efficient engineering of these often-huge biosynthetic machinery to produce novel bioactive NRPs is an ongoing challenge. Understanding the evolution of these multifunctional enzymes might provide new insight for engineering and discovering new peptide-based therapeutic agents[65]. A key feature of desotamide evolution that could be exploited for future NRPS engineering efforts is the intragenomic recombination between distinct NRPS encoding BGCs. The recombination event we predicted takes place within the boundaries of A domains, consistent with previous[39,40,66] and more

recent work[18,41]. More specifically, the predicted recombination allows for the substitution of the substrate specificity conferring the active site and the ATP/phosphate-binding catalytic P-loop[67] while functionally maintaining most of the A-T[68] and all of the C-A inter-domain contacts[69–71]. This pattern is frequently seen in our dataset, suggesting an evolutionary advantage for maintaining the structural relationships between the P-loop and substrate binding pocket. Our results are supported by two other independent studies, further highlighting its significance[18,41].

Calcott and co-workers analysis of recombination hotspots within C-A-T tri-domains (modules) from *Pseudomonas*, *Bacillus* and *Streptomyces* species identified the same recombination sites as predicted for the formation of *wolG1A2*[41]. Subsequently, Baunach and co-workers systematic in silico dissection of many individual recombination events unveiled the striking commonality of A domain recombination events in nature[18]. Specifically, these recombination events target variable portions of the $A_{core}$ domains to modulate A domain substrates while domain-domain interactions and the flexible $A_{sub}$ domain largely remained unaffected. These studies, taken together with the evolutionary and biochemical evidence presented here, must inevitably lead to a change in paradigm of future NRPS engineering experiments. Established engineering principles are being overhauled by the increasing evidence base. In particular, the idea that C and A domains have coevolved[72,73], resulting in strong acceptor site specificity of C domains[74,75] is in question[18,41,63]. This hypothesis is not congruent with the observation that A domain recombination is responsible for the diversification of many NRPs[18,39,76], not least the L-ornithine to glycine change demonstrated here. Such diversification would be impossible if C domain 'gatekeeping' was universal.

The ability to exchange A domains with greater accuracy should aid engineering efforts in the future, however many issues remain. As we have observed in nature, there is unlikely to be a single recombination site that will work in every case. Nevertheless, the growing body of genomic data will continue to reveal evolutionary snapshots akin to the wollamide BGC presented here. Over a decade of sequencing has provided us with only a threadbare sample of microbial genomic diversity and, as a result, we are still largely ignorant of the evolutionary processes that govern small molecule biosynthesis. Fortunately, this means there remains a wealth of information yet to be gleaned from Nature.

## Methods

**Strains and culture conditions**. All strains and plasmids used in the study can be found in Supplementary Tables 12 and 13. *Escherichia coli* strains were cultured on

LB medium (25 g/L LB broth (Miller)) at 37 °C. *Saccharomyces cerevisiae* CEN.PK 2-1c and derivatives were cultured on YPDA (10 g/L yeast extract, 20 g/L peptone and 20 g/L glucose, in deionised water) supplemented with 0.004% adenine sulfate. *Streptomyces* spp. were cultured on SF + M agar (20 g/L soya flour, 20 g/L mannitol, 20 g/L Lab M No 1 agar, in deionised water) at 30 °C or in SV2 liquid medium (glucose 15 g/L, glycerol 15 g/L, soy peptone 15 g/L, sodium chloride 3 g/L, calcium carbonate 1 g/L, in deionised water). Apramycin (100 μg/mL), kanamycin (50 μg/mL) and G418 (200 μg/mL) were used as selection markers.

*Streptomyces* sp. MST-70754, MST-71321, MST-71458, MST-94754 and MST-127221 were isolated by Microbial Screening Technologies (Smithfield, NSW, Australia) from soil samples collected across New South Wales between 1994 and 1998.

**Genome sequencing and genomic analysis.** High molecular weight genomic DNA was extracted according to a modified version the salting out procedure described by Kieser et al.[34] with the modifications as described here: wet mycelium (0.5 mL) from a 30 h old SV2 culture was washed with 10% sucrose (10 mL) before resuspension in SET buffer (5 mL; 75 mM NaCl, 25 mM EDTA, 20 mM TrisHCl pH 8.0) to which lysozyme (200 μL; 50 mg/mL) and ribonuclease A (15 μL; 10 mg/mL) were added. The cells were incubated overnight at 37 °C; fresh lysozyme (300 μL) was added after ca. 17 h followed by an additional 2 h incubation.

Genomic DNA of *Streptomyces* sp. MST-110588 was sequenced with Pacific Biosciences (PacBio) RSII SMRT technology (commissioned to the Earlham Institute Norwich, UK) and assembled via the HGAP2.0 pipeline. Genomic DNA of *Streptomyces* spp. MST-70754, MST-71321, MST-71458, MST-94754 and MST-127221 were sequenced with Illumina MiSeq using paired-end sequencing and Nextera Mate Pair library preparation and assembled in house (commissioned to the University of Cambridge DNA sequencing facility, Department of Biochemistry).

The genome sequence of *Streptomyces* sp. MST-110588 was deposited in GenBank with accession code CP074380. The DNA sequences of the desotamide BGCs were deposited at GenBank as follows: from *Streptomyces* sp. MST-70754 at accession code GenBank: MZ093610; MST-71321 at accession code GenBank: MZ093611; *Streptomyces* sp. MST-71458 at accession code GenBank: MZ093612; *Streptomyces* sp. MST-94754 at accession code GenBank: MZ093613; and *Streptomyces* sp. MST-127221 at accession code GenBank: MZ093614.

Genomic DNA sequences were annotated using prodigal[77] as implemented by antiSMASH 5.0[31]. Strain taxonomy was performed using multi-locus sequence typing was implemented in AutoMLST[78]. Adenylation-domain specificities were predicted by NRPSsp[79] or by NRPSpredictor2[80] and the specificity codes described by Stachelhaus[30] and Minowa[81], as implemented in antiSMASH. Gene homologues were identified from translated protein sequences using BLASTp[82]. Models of the assembly-line modules and adenylating domains were generated via remote homology detection using Phyre2[83] and visualised using PyMOL2.5[84].

**Metabolic analysis.** *Streptomyces* spp. were grown on SF + M agar for 14 days at 30 °C. 1 cm² agar plugs were taken from the plates and extracted by shaking with methanol (1 mL) for 30 min. Samples were centrifuged at 17,000 × g and the supernatant was removed under reduced pressure prior to resuspension in 100 μL methanol. Ultra-high performance liquid chromatography high-resolution mass spectrometry (UHPLC-HRMS) was performed on a Thermo Scientific Vanquish UHPLC system connected to a Thermo Scientific Q Exactive Orbitrap mass spectrometer in ES + and ES- modes. Samples were analysed using a Kinetex 1.7 μm C18 100 Å LC column, 50 ×2.1 mm (Phenomenex) with a flow rate of 0.5 mL/min and column compartment temperature of 40 °C. Solvents were A: Milli-Q water with 0.1% (v/v) formic acid (FA) and B: MeCN 0.1% (v/v) FA, with the following gradient: 5% to 95% B 0–6 min, 95% B 6–7.7 min, 95% to 5% B 7.7–7.9 min, 5% B 7.9–9 min. The flow was diverted to waste for the first 0.5 min. Nitrogen was used as the sheath gas. The injection volume was 10 μL for all samples. Data were acquired using Thermo Scientific Xcalibur 4.3 and analysed using Thermo Scientific Freestyle 1.6 software.

**Evolutionary analysis of NRPS assembly-lines.** Condensation domain and adenylation domain coding nucleotide sequences of *Streptomyces* sp. MST-110588, annotated by antiSMASH5.0, were aligned by ClustalW (cost matrix: BLOSUM, gap open cost: 10, gap extend cost: 0.1)[85]. Recombination detection was performed on the resulting adenylation-domain alignments using RDP[86], GENECONV[87], BOOTSCAN[88], MAXCHI[89], CHIMERA[90], SISCAN[91], LARD[92] and 3SEQ[93] as implemented in RDP4[44] using default settings. Breakpoints were plotted on translated protein sequences by manual annotation. Phylogenies were generated using FastTree 2[94].

**Assembly of the plasmids pBO1 and pBO1-wolG2.** Plasmid isolation from *E. coli* and *S. cerevisiae* was performed using PureYield Plasmid Miniprep System (Promega) and Zymoprep Yeast Plasmid Miniprep II (Zymo Research) respectively. The plasmid pBO1 is a selectable yeast centromeric-*E. coli*-*Streptomyces* shuttle plasmid. It is a derivative of the integrative *Streptomyces* expression vector pGP9[95]. The yeast cassette from pFF62A (KanMX4 and 2micron2 Ori)[96] was introduced into the pGP9 backbone to produce pBO1. pGP9 was linearised by PCR

amplification with Q5 High-Fidelity DNA polymerase (NEB) using primers pBO1-pGP9-F1 and pBO1-pGP9-R1, and the yeast cassette from pFF62A was amplified with primers pBO1-pFF-F1 and pBO1-pFF-R1 (Supplementary Table 14). The PCR linearised pGP9 and yeast cassette were transformed into *S. cerevisiae* CEN.PK 2-1c according to Schiestl and Gietz[97,98]. Plasmid pBO1-*wolG2* was generated by transformation associated homologous recombination (TAR). pBO1 was linearised with NdeI. *wolG2* was amplified with primers pBO1_wolG2_F1 and pBO1_wolG2_R1 (Supplementary Table 14) from gDNA of *Streptomyces* sp. MST-110588. Linear pBO1 and PCR amplified *wolG2* fragments were assembled by transformation into *S. cerevisiae* CEN.PK 2-1c. Plasmids were purified from G418 resistant colonies.

**Heterologous expression of wolG2 and metabolic analysis.** Plasmids pBO1 and pBO1-*wolG2* were extracted from *E. coli* DH5α using Wizard Plus SV miniprep system (Promega) and transformed into electrocompetent *E. coli* ET12567/pUZ8002[99]. The resulting strains were used for the conjugation. Conjugation of *Streptomyces* sp. MST-70754 spores was carried out according to Kieser et al.[34]. The resulting exconjugants were cultured on SF + M agar plates containing apramycin (100 μg/mL) and extracted with methanol and analysed by LCMS/MS as described above.

**Modelling of docking domain complexes.** Protein sequences of WolG1_NDD–WolH_CDD and WolG2_NDD–WolH_CDD as well as the crystal structure coordinates of PaxC_NDD–PaxB_CDD (retrieved from the RSCB Protein Data Bank file PDB-ID: 6TRP_1 [https://doi.org/10.2210/pdb6trp/pdb])[37] were loaded into the Molecular Operating Environment (MOE) 2019.0102[100]. Prior to homology modeling the crystal structure of 6TRP_1 was prepared (i.e. wrong protonation, chirality, and hybridisation) and a structural alignment was made. A series of 10 models per protein were constructed with MOE using a Boltzmann-weighted randomised procedure combined with specialised logic for the handling of sequence insertions and deletions[101,102]. The model with the best packing quality function was selected for full energy minimisation. Using the AMBER14 forcefield parameters for proteins (Amber14: EHT), the calculated MOE packing scores for models of WolG1_NDD–WolH_CDD and WolG2_NDD–WolH_CDD were 2.4286 and 2.2322, respectively. The stereochemical qualities of the models were assessed using Ramachandran plots and by calculating the Root-Mean-Square-Deviation (RMSD) values of the superposed Cα-atoms of the model with its respective template structure. RMSDs of WolG1_NDD–WolH_CDD and WolG2_NDD–WolH_CDD with 6TRP_1 were 0.1724 and 0.1762, respectively.

**Protein expression, purification and biochemical assays.** Adenylation domains were assembled into pET28a using the methodology described in section 1.5 using the primers from Supplementary Table 14. Additionally, the MbtH-like protein-coding sequence *wolF2* was cloned into pCDF-Duet-1 using the same procedure. *E. coli* 2(DE3)pLysS containing both expression plasmids were grown overnight in 10 mL LB. Overnight cultures were used to inoculate 1 L LB and grown to OD600 ~0.5, prior to induction with IPTG (1 mM). Following induction, cultures were grown overnight at 18 °C. Cultures were centrifuged at 8000 × g for 20 min and the supernatant was discarded. Pellets were resuspended in 15–20 mL P-Buffer (1 M K$_2$HPO$_4$ 9.4%; 1 M KH$_2$PO$_4$, 0.6%, pH 8.0). Cells were homogenised using an Avestin EmulsiFlex-B15 (40 bar) and the supernatant was removed and filtered.

Proteins were purified on an ÄKTA Pure chromatography system (GE Healthcare) using a His-Trap 1 mL column. The column was equilibrated with P-buffer until a stable UV signal was established. The equilibrated column was sequentially washed with 5 column volumes of P-buffer containing 10 mM, 20 mM and 50 mM imidazole. Proteins were eluted with 24 column volumes of P-buffer containing 250 mM imidazole. Fractions with an UV absorbance at 280 nm were pooled. The presence of the correct MW protein was assessed by SDS-PAGE using 12% RunBlue SDS protein gels (Expedeon). Protein concentration was determined by Bradford assay[103] (BIORAD).

Hydroxylamine trapping assays were performed using the method described by Kadi and Challis[48] with minor modifications. Enzyme (50 mM; 30 μL) was added to the assay solution for each amino acid (15 mM MgCl$_2$, 2.25 mM ATP, 150 mM hydroxylamine, 3 mM amino acid). Samples were incubated for 2 h prior to the addition of the stopping solution (10 % (w/v) FeCl$_3$·6H$_2$O, 3.3% trichloroacetic acid in 0.7 M HCl). The absorbance at 540 nm was measured using a BioMate 3 spectrophotometer (Thermospectronic).

**Reporting summary.** Further information on research design is available in the Nature Research Reporting Summary linked to this article.

## Data availability

Data supporting the findings of this work are available within the paper and its Supplementary Information file, or in publicly available databases. The DNA sequence data for the genome assembly and biosynthetic gene clusters (BGCs) generated in this study have been deposited in GenBank. The genome sequence of *Streptomyces* sp. MST-

110588 has been deposited under the accession code GenBank: CP074380. The desotamide BGCs have been deposited under the accession codes as follows: *Streptomyces* sp. MST-70754, GenBank: MZ093610; *Streptomyces* sp. MST-71321, GenBank: MZ093611; *Streptomyces* sp. MST-71458, GenBank: MZ093612; *Streptomyces* sp. MST-94754, GenBank: MZ093613; and *Streptomyces* sp. MST-127221, GenBank: MZ093614. The crystal structure coordinates of the docking domain protein PaxC_NDD–PaxB_CDD were retrieved from the RSCB Protein Data Bank file PDB-ID: 6TRP_1 [https://doi.org/10.2210/pdb6trp/pdb]. Biochemical data for analysis of adenylation-domain substrate specificity are available as a Source Data file. A reporting summary for this Article is available as a Supplementary Information file. Source data are provided with this paper.

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

## Acknowledgements

This work was supported by the Biotechnology and Biological Sciences Research Council (BBSRC) via Strategic Program Project BBS/E/J/000PR9790 to the John Innes Centre, and by Norwich Research Park Doctoral Training Program Studentship BB/J014524/1 to T.J.B.) and BB/M011216/1 (to J.D.L.). We also acknowledge the support of the Cooperative Research Centres Projects Scheme (CRCPFIVE000119) (to E.L.). We thank the Earlham Institute (Norwich UK) for sequencing and assembly of the *Streptomyces* sp. MST110588 genome. We also thank the JIC metabolomics platform at for excellent mass spectrometry support.

## Author contributions

B.W. and E.L. conceived the study. B.W., T.J.B., K.A.J.B. and J.L. designed the experiments, and T.J.B., K.A.J.B., J.L. and S.F.D.B. performed the experiments and analysed the data. B.W. and T.J.B. wrote the paper and all authors reviewed and edited the manuscript.

## Competing interests

E.L. is a co-owner of Microbial Screening Technologies. The remaining authors declare no competing interests.
