## [Peer Review File · Nature Communications]

Bifurcation drives the evolution of assembly-line biosynthesisREVIEWER COMMENTS

Reviewer #1 (Remarks to the Author):

I very much enjoyed reading the paper by Booth, Bozhüyük et al. It is very well written, provides exciting new insights into NRPS evolution, and the conclusions are strongly supported by the data. The bioinformatic analyses are well done and the experimental work looks equally solid. I think this will be a landmark paper on this topic for years to come. It has profound implications not only for the field of natural product enzymology, but also for molecular evolution and biosynthetic engineering.

I only have a couple of minor comments:

- Line 118: 'Technologies' (add apostrophe or add 'the' before 'Microbial?')
- Line 119: desotamide BGCs (plural, as multiple strains are concerned)
- Line 144: add comma after 'methods'
- Line 180: I do not think 'speciation' is the right term here. Perhaps 'sequence divergence', or in combination with 'adaptation' or 'directional selection'?
- Line 197: 'adenylation-domain-encoding' (hyphens)
- Line 259: 'WolH'
- Line 263: 'code for' or 'encode', not 'encode for'
- Line 296: add comma after 'MST110588'
- Line 328: I realize this came out after submission of this paper, but it might be nice to cite the GRINS paper by Nivina et al. (PNAS, 2021) here and perhaps add some brief discussion.
- Figure 4a: was there a particular reason to use fasttree2 instead of e.g. RAxML or IQ-TREE here for such a small set of sequences? I do not think it will matter for the results, but I am just curious.
- Figure 4c: please indicate the sliding window size in the figure legend.
- Figure 5 legend: 'colour-coded' (add hyphen)

Reviewer #2 (Remarks to the Author):

In this work, the authors analyzed a set of present-day nonribosomal peptide synthetase clusters to reconstitute their evolutionary history and demonstrate how gene duplication and neofunctionalization can result in biosynthetic pathway bifurcation. The combination of sequence analysis, biochemical assays, and engineering approaches builds a strong case in support of the proposed model of biosynthetic assembly-line enzyme evolution (Fig. 6). The manuscript is easy to follow, and the discussion nicely puts the results into the context of both fundamental understanding of biosynthetic pathway evolution and the practical aspects of their engineering. This work will be of high significance for the field of natural products.

There are some points in the manuscript that warrant correction or clarification.

1. In the Abstract, the authors claim that "neofunctionalization occurs primarily through intragenomic recombination". While this work clearly shows that intragenomic recombination is an important mechanism for neofunctionalization, further analysis would be needed before it can be declared as the predominant one.
2. The term "neofunctionalization" does not (yet) have a universal meaning, so it would be good to define it in the first instance of use
2. References to Fig.1a and Fig.1b are swapped (p.3).
3. In Supplementary Fig.1, clusters from only four out of five new desotamide producers are shown (cluster from MST-71458 is absent). Instead, two previously described clusters are shown: those synthesizing ulleungmycin and dechlorocuracomycin. Just like desotamides, these compounds are similar to wollamides, but have several key structural differences. This warrants their mention in the main text, and perhaps a short discussion about their place in the

evolutionary picture.

4. It is not clear from the description of the engineering approach (p.5 of main text and/or section 1.6 of Supplementary Information) that pBO1 is an integrative vector.

5. Typo: "and" missing between 74.3% and 71.2%.

6. In Fig.4c, it seems that the colors of the two lines (red and blue) should be swapped. Also, yellow line is not very visible; a different color would be helpful.

7. The legends to Fig.4 a, b and c do not clearly mention that DNA - not protein - sequences were used. Similarly, the legend to Supplementary Fig.6 does not mention that protein sequence was used in this case.

8. Typo (p.8): WolH instead of WollH.

9. It is not clear from the text that a single intragenomic recombination event could not account for the type of sequence exchanges observed in adenylation domains. Clarification is needed that such an exchange would require either a double crossover, or a gene conversion event (e.g. on p.9).

10. On p.9, the intermediate NRPS that resulted after gene duplication and recombination is described to be "selective for glycine". Would not the ancestral and the recombined copy of WolG homologue in that cluster be selective of the ancestral substrate and glycine, respectively?

11. One additional point that the authors might want to address in the discussion is the SAR of wollamides and desotamides, described in "Structure-Activity Relationships of Wollamide Cyclic Hexapeptides with Activity against Drug-Resistant and Intracellular Mycobacterium tuberculosis" by Khalil et al. One would naively assume that compounds with less antimicrobial activity (at least against human pathogens) would have been more ancestral, while additional activities would arise during evolution. It appears that in this case however, the evolutionary path has led from wollamides which show significant antimycobacterial activity (in addition to their activity against Gram-positive pathogens) towards desotamides which have lost this activity.

12. Fig.6 contains a new term "modular collapse", which is not introduced in the text or in the legend

Reviewer #3 (Remarks to the Author):

Comments to the Authors

NRPS pathways give rise to an astonishing diversity of natural products. However, how this immense variety has evolved is still a matter of debate. In the current manuscript, the authors unravel the genetic and enzymatic basis for the concurrent production of wollamide and desotamide congeners in a single strain. Based on sequence comparison the authors convincingly concluded that duplication of a NRPS coding gene from an ancestral wollamide pathway and neofunctionalization of the duplicated version by intragenomic recombination led to pathway bifurcation. Moreover, the authors speculate that depreciation and eventual loss of the original gene created the new linear desotamide pathway, which is conserved among various *Streptomyces* spp. Based on this "evolutionary snapshot" the authors introduce a more general model that explains how whole gene duplication and neofunctionalization can propel the diversification of NRPS pathways.

The authors' approach is logical and the manuscript is well-organized and – predominantly (see below) – well-written. The topic is of relevance to a broad readership and it will make a nice addition to the journal. However, I would not recommend publication without the following major changes/additions:

One of the key experiments of this study is the successful engineering of wollamide production in a desotamide-only producing strain. However, the authors fail to provide convincing evidence for the production of wollamides. Although the authors state that "presence of both wollamide and desotamide congeners was confirmed for the engineered strains by comparison of retention time,

isotopic masses and MS/MS fragmentation of the compounds produced by the native wollamide producer (Fig. 3, Supplementary Fig. 3).” Fig. 3 only provides extracted ion chromatograms without comparison to authentic, NMR-proven standards. Moreover, also the SI lacks comparison to authentic standards and no MS/MS spectra comparison is provided. This would be the minimum requirement to judge on the plausibility of wollamide production, especially since – like in this case – only traces are produced. On top of that, retention times of wollamide A in Fig. 3 and Supplementary Fig. 3, clearly deviate (see picture attached), making the evaluation of wollamide A production even less convincing...

Moreover, one fundamental aspect of the evolutionary model is the depreciation of the original phenotype, which marks the transition to linear pathways. The authors speculate, that a mutation in the docking domain in the original version most likely is causing decreased docking domain pair affinities and that this could explain the low relative yields of wollamides in the engineered strain. This is an interesting theory, but it should be backed up by experimental evidence. Introducing point mutations (even in large plasmids) nowadays is an easy standard technique and should be done to investigate the *in silico* results. Moreover, improved wollamide production due to a mutated docking domain could help to convincingly confirm the production of wollamide A in the engineered strain. Improved production after docking domain mutation would also help to rule out, that impaired wollamide production is a consequence of the missing MbtH-like protein WolF2 in the engineered strain (an option which isn't discussed by the authors). There might be a good reason why its coding gene is co-duplicated and the fact, that only co-expression with the MbtH-like protein WolF2 yielded soluble protein *in vitro* strongly hints to an important role as well..

Minor points:

Some sentences are strange/hard to understand.

For example line 185-188

“More specifically, given the reduced rate of horizontal gene transfer between distantly related taxa and the high rate of heterogeneity between recombinant sequences has, it has been speculated that intragenomic recombination within ancestral strains can provide opportunities for assembly-line diversification 18,42.”

Line 298-300

“This observation suggested that the deprecation of interactions between the C- and N-terminal docking domains of WolH/DsaH and WolG2 respectively.

RESPONSE TO REVIEWER COMMENTS

Reviewer #1 (Remarks to the Author):

I very much enjoyed reading the paper by Booth, Bozhüyük et al. It is very well written, provides
exciting new insights into NRPS evolution, and the conclusions are strongly supported by the data.
The bioinformatic analyses are well done and the experimental work looks equally solid. I think this
will be a landmark paper on this topic for years to come. It has profound implications not only for the
field of natural product enzymology, but also for molecular evolution and biosynthetic engineering.

**We thank reviewer #1 for their kind remarks and have addressed all the minor comments below.**

I only have a couple of minor comments:

• Line 118: Technologies' (add apostrophe or add 'the' before 'Microbial'?)

**Done.**

• Line 119: desotamide BGCs (plural, as multiple strains are concerned)

**Done.**

• Line 144: add comma after 'methods'

**Done.**

• Line 180: I do not think 'speciation' is the right term here. Perhaps 'sequence divergence', or in
combination with 'adaptation' or 'directional selection'?

**We have changed this sentence to read: 'Given the overall similarity of these genes, we deemed it
unlikely that such high sequence variation could emerge through the accumulation of point
mutations alone.'**

• Line 197: 'adenylation-domain-encoding' (hyphens)

**Done.**

• Line 259: 'WolH'

**This should read 'WollH' as it is signifying the first two proteins in the assembly-line.**

• Line 263: 'code for' or 'encode', not 'encode for'

**Changed to: '...which encode the selection and activation of glycine and L-ornithine respectively.'**

• Line 296: add comma after 'MST110588'

**Done.**

• Line 328: I realize this came out after submission of this paper, but it might be nice to cite the GRINS
paper by Nivina et al. (PNAS, 2021) here and perhaps add some brief discussion.

**Nucleotide skew provides additional evidence for gene duplication and so we have added reference
to this. See: "Additionally, *woI*G1, *woI*G2 and *dsa*G share common patterns of nucleotide skew,
a phenomenon indicative of a recent divergence."**

**We also amended Supplementary Figure 6 to include graphs of nucleotide skew. We also amended
the discussion to read: "Comparison of the protein sequences of the NRPSs encoded by *woI*G1/*G*2
indicated a high degree of sequence conservation and identical GC-skews, indicative of a gene
duplication event." GRINS are a fascinating discovery, and we would love to discuss more in the
future, but given the constraints of a communication, we are unable to expand further than we have
done already.**

• Figure 4a: was there a particular reason to use fasttree2 instead of e.g. RAxML or IQ-TREE here for
such a small set of sequences? I do not think it will matter for the results, but I am just curious.

**We have run the analysis also using RAxML and the results are functionally the same. The major
difference being that RAxML requires an outgroup and cannot produce unrooted trees. Given that we
wanted the figure to best represent the output of RDP, we settled on fasttree. As for IQ-Tree, it is just
not part of our usual pipeline.**

• Figure 4c: please indicate the sliding window size in the figure legend.

**Done**

• Figure 5 legend: 'colour-coded' (add hyphen)

**Done**

Reviewer #2 (Remarks to the Author):

In this work, the authors analyzed a set of present-day nonribosomal peptide synthetase clusters to
reconstitute their evolutionary history and demonstrate how gene duplication and neofunctionalization
can result in biosynthetic pathway bifurcation. The combination of sequence analysis, biochemical
assays, and engineering approaches builds a strong case in support of the proposed model of
biosynthetic assembly-line enzyme evolution (Fig. 6). The manuscript is easy to follow, and the
discussion nicely puts the results into the context of both fundamental understanding of biosynthetic
pathway evolution and the practical aspects of their engineering. This work will be of high significance
for the field of natural products.

**We thank reviewer #2 for their kind remarks and have addressed all the minor comments below.**

There are some points in the manuscript that warrant correction or clarification.

1. In the Abstract, the authors claim that "neofunctionalization occurs primarily through intragenomic
recombination". While this work clearly shows that intragenomic recombination is an important
mechanism for neofunctionalization, further analysis would be needed before it can be declared as
the predominant one.

**We use 'primarily' to reflect the fact that other mechanisms to highlight that other mechanisms (point
mutation etc.) are required in this process following recombination, but we understand that the
wording is ambiguous. We have rephrased to: 'We show that, in the case of the wollamide
biosynthesis, neofunctionalisation is initiated by intragenomic recombination.'**

2. The term "neofunctionalization" does not (yet) have a universal meaning, so it would be good to
define it in the first instance of use.

**We understand the term to mean the acquisition of new function following a gene duplication event
and have amended the text in the discussion to provide this context: "In this fashion, a duplicate may
evolve new functionality (neofunctionalization) without losing the original product."**

2. References to Fig.1a and Fig.1b are swapped (p.3).

**Fixed. Also made an additional reference to 1a to conserve the order in the text.**

3. In Supplementary Fig.1, clusters from only four out of five new desotamide producers are shown
(cluster from MST-71458 is absent). Instead, two previously described clusters are shown: those
synthesizing ulleungmycin and dechlorocuracomycin. Just like desotamides, these compounds are
similar to wollamides, but have several key structural differences. This warrants their mention in the
main text, and perhaps a short discussion about their place in the evolutionary picture.

**We have corrected the figure and added the following lines to the discussion for context:**

**"A growing number of hexapeptide BGCs are becoming available, for example those of the
related ulleungmycin and curacomycin. Given the small yet diverse differences of the peptide
backbones, the hexapeptides present an enticing model for understanding NRPS evolution."**

4. It is not clear from the description of the engineering approach (p.5 of main text and/or section 1.6
of Supplementary Information) that pBO1 is an integrative vector.

**We have amended the section in the main text to read: 'an integrative Streptomyces expression
vector.' We amended the SI to read: "It is a derivative of the integrative Streptomyces expression
vector pGP9." This information is also present in Supplementary Table 13.**

5. Typo: "and" missing between 74.3% and 71.2%.

**Fixed.**

6. In Fig.4c, it seems that the colors of the two lines (red and blue) should be swapped. Also, yellow
line is not very visible; a different color would be helpful.

**The colours are the correct way around. We agree that yellow is not the best choice of colour so we
have changed this to orange. It is meant to be less visible than the other two lines as this comparison
is less important for the discussion.**

7. The legends to Fig.4 a, b and c do not clearly mention that DNA - not protein - sequences were
used. Similarly, the legend to Supplementary Fig.6 does not mention that protein sequence was used
in this case.

**Fixed**

8. Typo (p.8): WolH instead of WollH.

**Done**

9. It is not clear from the text that a single intragenomic recombination event could not account for the
type of sequence exchanges observed in adenylation domains. Clarification is needed that such an

exchange would require either a double crossover, or a gene conversion event (e.g. on p.9).
We agree that the terminology could be considered a little confusing. However, the exact mechanism
is unclear. Most papers on this topic do not discuss this either. However, for clarity we have added in
the following text on page 6: 'e.g. the exchange of alleles via a double recombination event'.

10. On p.9, the intermediate NRPS that resulted after gene duplication and recombination is
described to be "selective for glycine". Would not the ancestral and the recombined copy of WolG
homologue in that cluster be selective of the ancestral substrate and glycine, respectively?

Yes. We were inferring glycine *and* ornithine, but the wording can be improved. This confusion is
somewhat compounded by the following sentence, so we have reworded to:

"Subsequently, an intragenomic recombination event between the DNA encoding the adenylation
domains of the duplicated module six in the *wolG* homologue and *orf6595* resulted in an intermediate
NRPS selective for **glycine** (leading to WolG1) **and L-ornithine** (leading to WolG2). This progenitor is
the common ancestor of the wollamide and desotamide producing *wol*/BGC sequenced here. **In one**
**lineage**, selection and mutation resulted in the *wol*/BGC observed in the contemporary genome of
*Streptomyces* sp. MST110588 capable of bifurcated biosynthesis. In a divergent lineage, the
ancestral gene encoding the L-ornithine-specific adenylation domain along with the duplicated MbthH-
like protein encoding gene and associated genes encoding L-ornithine biosynthesis were lost through
gene deletion, ultimately resulting in the contemporary *dsa* BGCs producing only desotamides."

11. One additional point that the authors might want to address in the discussion is the SAR of
wollamides and desotamides, described in "Structure-Activity Relationships of Wollamide Cyclic
Hexapeptides with Activity against Drug-Resistant and Intracellular Mycobacterium tuberculosis" by
Khalil et al. One would naively assume that compounds with less antimicrobial activity (at least
against human pathogens) would have been more ancestral, while additional activities would arise
during evolution. It appears that in this case however, the evolutionary path has led from wollamides
which show significant antimycobacterial activity (in addition to their activity against Gram-positive
pathogens) towards desotamides which have lost this activity.

We considered this line of reasoning also, however we decided to omit this from the discussion for
several reasons. The primary reason is that we have no knowledge of the ecology of the strain or the
molecules themselves. As such, we can't meaningfully comment on this observation. It is possible
that the evolution of the desotamides and the wollamides has nothing to do with antimycobacterial
activity. Alternatively, within some ecological niche it might have been beneficial to maintain
*Mycobacterial* neighbours while inhibiting/eliminating other competitors. Given that the most active
synthetic wollamide analogue tested is not seen in Nature we can infer that it is not the sole driving
force. Nevertheless, we do believe Khalil et al's work is of significant importance to field and we
should have included this reference previously so have added it to the references in the introduction.

12. Fig.6 contains a new term "modular collapse", which is not introduced in the text or in the legend.
We agree that this terminology is undefined. After consideration, we decided to use the more
accepted term of 'modular deletion.'

Reviewer #3 (Remarks to the Author):

Comments to the Authors

NRPS pathways give rise to an astonishing diversity of natural products. However, how this immense
variety has evolved is still a matter of debate. In the current manuscript, the authors unravel the
genetic and enzymatic basis for the concurrent production of wollamide and desotamide congeners in
a single strain. Based on sequence comparison the authors convincingly concluded that duplication of
a NRPS coding gene from an ancestral wollamide pathway and neofunctionalization of the duplicated
version by intragenomic recombination led to pathway bifurcation. Moreover, the authors speculate
that depreciation and eventual loss of the original gene created the new linear desotamide pathway,
which is conserved among various *Streptomyces* spp. Based on this "evolutionary snapshot" the
authors introduce a more general model that explains how whole gene duplication and
neofunctionalization can propel the diversification of NRPS pathways.

The authors' approach is logical and the manuscript is well-organized and – predominantly (see

below) – well-written. The topic is of relevance to a broad readership and it will make a nice addition
to the journal. However, I would not recommend publication without the following major
changes/additions:

We thank reviewer #3 for their kind remarks and have addressed all the major and minor comments
below.

One of the key experiments of this study is the successful engineering of wollamide production in a
desotamide-only producing strain. However, the authors fail to provide convincing evidence for the
production of wollamides. Although the authors state that “presence of both wollamide and
desotamide congeners was confirmed for the engineered strains by comparison of retention time,
isotopic masses and MS/MS fragmentation of the compounds produced by the native wollamide
producer (Fig. 3, Supplementary Fig. 3).” Fig. 3 only provides extracted ion chromatograms without
comparison to authentic, NMR-proven standards. Moreover, also the SI lacks comparison to authentic
standards and no MS/MS spectra comparison is provided. This would be the minimum requirement to
judge on the plausibility of wollamide production, especially since – like in this case – only traces are
produced. On top of that, retention times of wollamide A in Fig. 3 and Supplementary Fig. 3, clearly
deviate (see picture attached), making the evaluation of wollamide A production even less
convincing...

We agree that there is ambiguity in the data presented due to experiments being run at different times
and on different machines. To address this issue, and the request for comparison to an authentic
standard, we have rerun the MS experiments to include authentic standards and comparisons to the
wild type strains. We have changed Figure 3 to show the new data and have added additional
supplementary figures (3-5) and amended other numbers respectively. We have also updated the
methodology in the supplementary information in line with the new experiments.

Moreover, one fundamental aspect of the evolutionary model is the depreciation of the original
phenotype, which marks the transition to linear pathways. The authors speculate, that a mutation in
the docking domain in the original version most likely is causing decreased docking domain pair
affinities and that this could explain the low relative yields of wollamides in the engineered strain. This
is an interesting theory, but it should be backed up by experimental evidence. Introducing point
mutations (even in large plasmids) nowadays is an easy standard technique and should be done to
investigate the in silico results. Moreover, improved wollamide production due to a mutated docking
domain could help to convincingly confirm the production of wollamide A in the engineered strain.
Improved production after docking domain mutation would also help to rule out, that impaired
wollamide production is a consequence of the missing MbtH-like protein WolF2 in the engineered
strain (an option which isn't discussed by the authors). There might be a good reason why its coding
gene is co-duplicated and the fact, that only co-expression with the MbtH-like protein WolF2 yielded
soluble protein in vitro strongly hints to an important role as well..

The proposed depreciation experiments are an excellent idea and indeed have been written into
funding proposals under consideration. Moreover, we do not currently have an individual in the lab to
carry out these experiments. We feel that the current analysis is sufficient for this story as the
depreciation narrative is supportive yet not essential for our arguments. We plan to work on this area
going forward.

We further appreciate the comments regarding the MbtH-like gene product WolF2 being required for
the function of the heterologous NRPS WolG2 in the engineered strain. However, we do not believe
this is a significant issue for a number of reasons. Chiefly, MbtH domains are not generally known to
function specifically in this fashion. It has been shown previously that a single copy of an MbtH
domain is sufficient for adenylation domain function. Indeed, in our own example WolF2 promoted the
solubility of all three A domain proteins, not just WolG2A2. Additionally, as we note in the manuscript,
the relative titres of wollamides are much lower than desotamides in the native producer also, so it is
not unsurprising to that titres are low in the engineered strain. Overall, we believe that depreciation as
described is by far the most likely explanation of the low titres of wollamide production in the hybrid
strain. However, we do concede that there is room for other explanations and have amended the
discussion to accommodate for such possibilities: ***“Although DD depreciation is likely to account for***

*the difference in yield between the wollamides and desotamides in WT MST-110588, other factors*
*may account for further loss of titre in our engineered strains, such as differences in the Mbth*
*domains or the reduced capacity to biosynthesise ornithine.”*

Minor points:

Some sentences are strange/hard to understand.

For example line 185-188

“More specifically, given the reduced rate of horizontal gene transfer between distantly related taxa
and the high rate of heterogeneity between recombinant sequences has, it has been speculated that
intragenomic recombination within ancestral strains can provide opportunities for assembly-line
diversification 18,42.”

Amended to: “the high rate of heterogeneity between recombinant **sequences, it has been**
**speculated....”**

Line 298-300

“This observation suggested that the deprecation of interactions between the C- and N-terminal
docking domains of WolH/DsaH and WolG2 respectively.

Amended to: “This observation suggested that the interactions between the C- and N-terminal
docking domains of WolH/DsaH and WolG2 **had depreciated.”**

We have also proofread the paper again to catch additional poorly worded sentences.

REVIEWERS' COMMENTS

Reviewer #2 (Remarks to the Author):

The revised manuscript is suitable for publication

Reviewer #3 (Remarks to the Author):

The authors have very thoroughly addressed all concerns about data presentation. Now the results appear very convincing.

I would have liked to see additional experiments to support the depreciation hypothesis. However, I understand that this is beyond the scope of the current manuscript. The addition of alternative explanations for the low wollamide A titers in the manipulated strain is a good compromise in my opinion. All in all, I fully support the publication of the revised manuscript.